# Monsters in the Dark: Sanitizing Hidden Threats with Diffusion Models

## Abstract

Steganography is the art of hiding information in plain sight. This form of covert communication can be used by bad actors to propagate malware, exfiltrate victim data, and communicate with other bad actors. Current image steganography defenses rely upon steganalysis, or the detection of hidden messages. These methods, however, are non-blind as they require information about known steganography techniques and are easily bypassed. Recent work has instead focused on a defense mechanism known as sanitization, which eliminates hidden information from images. In this work, we introduce a novel blind deep learning steganography sanitization method that utilizes a diffusion model framework to sanitize universal and dependent steganography (DM-SUDS), which both sanitizes and preserves image quality. We evaluate this approach against state-of-the-art deep learning sanitization frameworks and provide further detailed analysis through an ablation study. DM-SUDS outperforms previous sanitization methods and improves image preservation MSE by 71.32%, PSNR by 22.43% and SSIM by 17.30%. This is the first blind deep learning image secret sanitization framework to meet these image quality results.

## 1 Introduction

Steganography, or the art of hiding information in plain sight, is a means of covert communication used commonly throughout history. In steganography, a secret message is embedded in a seemingly harmless physical or digital medium, termed a cover. This combined message (secret + cover = container) is then relayed to an intended recipient. Due to the hidden nature of the secret message, communication can occur between parties without arousing suspicion as to the true intent of the dispatched container.

This process can be applied to both physical mediums (e.g., microdots, invisible ink, embedded objects, etc.) as well as digital mediums, including images, videos, audio, and text. The pervasive nature of digital media presents a heightened risk for steganography. Its widespread accessibility and ease of dissemination make digital platforms more susceptible to covert manipulations, potentially enabling malicious actors to spread concealed information or malware at an unprecedented scale and speed. In addition to the widespread effect of malware propagation, steganography via digital mediums can also be used to exfiltrate victim data and communicate with other bad actors. As such, advanced detection and prevention mechanisms are necessary to combat against these harmful use cases. While various digital mediums can be used as covers, we focus on the use of images in this work, as these are more accessible and commonly used in the wild.

Current image steganography defenses utilize a technique known as steganalysis, or the detection of hidden messages in images Johnson & Jajodia (1998b); Bachrach & Shih (2011). Steganalysis tools analyze images for known signatures and/or anomalies in pixel values, noise distributions, and other statistical measures to indicate the presence of steganographic content. Recent work has also incorporated machine learning strategies to enhance detection accuracy Zhang et al. (2018); Xu et al. (2016); Xu (2017); Ye et al. (2017); Qian et al. (2018). These defense strategies, however, rely upon data curated from preexisting steganographic techniques and are referred to as *non-blind*. While adept at detecting known steganography, these defense mechanisms are useless against new forms of steganography in the wild – especially ones engineered to bypass these existing systems Li et al. (2022); Hayes & Danezis (2017); Tang et al. (2019).

Another defense mechanism against steganography is sanitization, which eliminates the presence of any potential secret message while maintaining the integrity of the cover media. For instance, a picture of a dog containing an executable malware binary would be sanitized if the malware binary is removed and the original image is minimally changed. The authors in Robinette et al. (2023) demonstrate the success of such an approach, utilizing a variational autoencoder strategy termed SUDS. The authors show that SUDS is able to protect against least significant bit (LSB), dependent deep hiding (DDH), and universal deep hiding (UDH) steganography, each of which is an unseen method by the sanitizer prior to testing (*blind*). Sanitization, therefore, is able to blindly protect systems and is a more robust defense strategy compared to detection.

While SUDS is able to successfully sanitize secrets, its ability to reconstruct the original image deteriorates as image complexity increases, which is a result of utilizing a variational autoencoder approach. To address the limited reconstruction capabilities of SUDS while maintaining sanitization performance, we propose a **d**iffusion **m**odel approach to **SUDS**, termed DM-SUDS. While diffusion models are most commonly used for their generative properties, they are trained as denoising mechanisms. By using a diffusion model to denoise potentially steganographic images, we believe that this approach will provide an improved alternative to SUDS, advancing the state-of-the-art in blind deep learning sanitization techniques for image steganography while increasing the potential breadth of its impact. All code to reproduce experiments is available at: `https://anonymous.4open.science/r/dmsuds-1D7C/README.md`. The contributions of this work, therefore, are the following:

1. **Implementation of a Novel Sanitization Framework:** We introduce a novel blind deep learning steganography sanitization method that utilizes a diffusion model framework to sanitize universal and dependent steganography.

2. **Demonstration of Sanitizer Capabilities:** We compare this approach with the current state-of-the-art sanitization method (SUDS) and demonstrate a 71.32% (MSE), 22.43% (PSNR), and 17.30% (SSIM) improvement in recovered image quality while maintaining sanitization performance.

3. **Ablation Study:** We further analyze this approach by conducting an ablation study on the model framework and evaluating sanitization performance on the ImageNet dataset.

## 2 BACKGROUND

In this section, we introduce steganography, existing sanitization techniques, and the metrics used for evaluation. While this work is medium agnostic, we focus on images represented by a matrix ($c$, $h$, $w$), where $c$ is the number of color channels, $h$ is the height, and $w$ is the width of the image.

### 2.1 STEGANOGRAPHY

**Notation** As shown by the *Pre-Sanitization* section in figure 1, steganography typically consists of a *cover*, *secret*, *container*, and a *revealed secret*. A *secret* $S$ is hidden within a *cover* $C$ using a hide function $\mathcal{H}$ to create a *container* $C'$ such that the difference between the cover and the container is minimal, or $\mathcal{H}(C, S) = C' \mid \text{MSE}(C, C') \to 0$. The *revealed secret* $S'$ can then be obtained from the container using a reveal function $\mathcal{R}$, which is usually the inverse of $\mathcal{H}$. The revealed secret should be minimally different from the original secret hidden in the cover, or $\mathcal{R}(C') = S' \mid \text{MSE}(S, S') \to 0$. In regard to sanitization, an image is sanitized with a sanitization function $\mathcal{P}$ to create a *sanitized image* $\hat{C}$, or $\mathcal{P}(X) = \hat{C} \mid X \in \{C, C'\}$. An attempted revealed secret from a sanitized image is denoted as $\hat{S}$, or $\mathcal{R}(\hat{C}) = \hat{S}$. In this work, $\mathcal{P} \in \{\text{SUDS}, \text{DM-SUDS}\}$, where DM-SUDS is the introduced diffusion model approach.

**Hiding Methods** There are many types of hiding techniques in steganography, including traditional and deep hiding. Traditional hiding involves either 1) modifying the pixels of the image (*spatial domain*) or 2) altering the image using a frequency distribution (*transform domain*) Cheddad et al. (2010); Subhedar & Mankar (2014); Trivedi et al. (2016); Kishor et al. (2016); Johnson & Jajodia (1998a). While information loss is inevitable, the user must strike a balance between maximizing the secret retained in the container and avoiding detection, hiding capacity vs. invisibility. With a higher hiding capacity, a secret is more likely to be detected in a container, but with

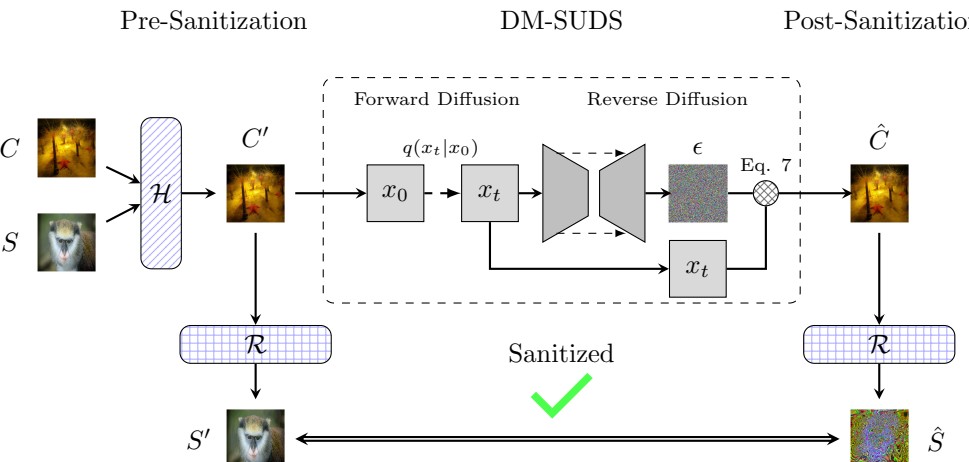

Figure 1: DM-SUDS (center) takes as input a cover or a container image $x_0 \in \{C, C'\}$ created with any type of steganographic technique. A noisy image is then sampled from this image at timestep $t$. In the reverse diffusion process, a denoising U-Net is used to predict the amount of noise added to the image, which is then used to recover the original image, resulting in a sanitized image $\hat{C}$. In the pre-sanitization phase, the secret is recoverable, as demonstrated in the bottom-left of the figure $S'$. After sanitization with DM-SUDS, however, a secret is not recoverable, as indicated by the bottom-right of the figure $\hat{S}$. This image, therefore, is successfully sanitized.

a lower invisibility, less information is transferable. Traditional methods are marred by this trade-off. In deep hiding, however, methods are able to incorporate a high capacity of the secret while maintaining invisibility, as these methods are capable of maintaining the pixel distributions of the cover image. Deep hiding techniques utilize deep neural networks as both the hide and reveal functions and fall into two main categories: dependent deep hiding (DDH) Volkhonskiy et al. (2020); Yang et al. (2019); Tang et al. (2017; 2020); Wu et al. (2020); Zhu et al. (2018); Wang et al. (2018); Baluja (2017) and universal deep hiding (UDH) Zhang et al. (2020). In DDH, the resulting container is cover dependent, and in UDH, the secret can be combined with any cover. In this work, we utilize a method from each of these categories 1) traditional = least significant bit (LSB) method Kurak & McHugh (1992), 2) dependent deep = DDH, and 3) universal deep = UDH. The implementations for DDH and UDH are CNN-based implementations from the same code base[1], and the LSB implementation is from the SUDS codebase[2]. Please see the supplementary material for more information on these methods.

## 2.2 SANITIZATION

**Traditional Sanitization** Sanitization, or active steganalysis, is the process of removing hidden information from potentially steganographic images while preserving the image quality of the cover. Traditional sanitization approaches attempt to remove hidden information by modifying the pixels of the image or the frequency distribution of the image, much like traditional steganographic approaches. Examples include flipping the bits of N LSB planes Paul & Mukherjee (2010), adding Gaussian noise to the image, and applying image filters Ameen & Al-Badrany (2013); Amritha et al. (2019); Geetha et al. (2021). While traditional sanitization techniques are effective on traditional hiding algorithms, they largely degrade the image quality of the restored image and are ineffective on deep hiding techniques.

**Deep Learning Sanitization** Deep learning sanitization attempts to improve upon the limited capabilities of traditional sanitization, especially in regard to deep learning steganography. In Jung et al. (2021), the authors present PixelSteganalysis which uses an 'analyzer' to predict pixel and edge distributions of a container, and an 'eraser' to sterilize the images by adjusting suspicious

---

[1] **DDH/UDH:** https://github.com/ChaoningZhang/Universal-Deep-Hiding
[2] **SUDS Code:** https://github.com/pkrobinette/suds-ecai-2023

pixels indicated by the analyzer. Their approach is extremely inefficient and also requires known information about common hiding areas from steganographic techniques to train the edge distribution model. In Wei et al. (2022), the authors use a median filter as a secret data remover (SDR). While the data removal process is blind, the image fusion and latent removal components utilize steganographic images embedded with text-secrets in the training process. Convolutional neural network (CNN) based approaches are introduced in Zhu et al. (2021), Zhu et al. (2022), and Hatoum et al. (2021) to sanitize steganography and/or watermarked images. Each of these methods, however, only considers text-based secrets.

To further enhance the resulting image quality of a sanitized image, some approaches have used generative models. In Corley et al. (2019), the authors apply a generative adversarial network (GAN) based approach called Deep Digital Steganography Purifier (DDSP). Their framework is only tested on traditional hiding methods and only considers text-based malware payloads as secrets. Additionally, DDSP requires previous knowledge of steganographic techniques in the training process and is, therefore, a non-blind approach. In Li et al. (2021), the authors present another GAN based approach, where a U-Net generator is trained to attack watermarked images. This approach is non-blind as steganographic images are used in the training process. The authors in Zuppelli et al. (2021) introduce a variational autoencoder (VAE) sanitizer, which is tested by sanitizing images embedded with malware powershell scripts via Invoke-PSImage[3]. The approach is evaluated by an open-source detection tool for LSB, StegExpose, which has comparable results to random guessing Baluja (2017). While the authors introduce VAE sanitization, they only evaluate LSB text embeddings, which are more fragile than image embeddings as small perturbations can drastically affect text semantics. The authors in Robinette et al. (2023) also utilize a variational autoencoder approach in a framework called SUDS but provide a more thorough analysis of this approach, including comparisons to traditional sanitization (Gaussian noise), incorporating more robust steganography techniques (LSB, DDH, UDH), and evaluating performance on encoded image secrets rather than text secrets. As SUDS is the most applicable to our introduced implementation, we provide a direct comparison to their work and utilize similar evaluation techniques.

## 2.3 Image Metrics

The metrics used to evaluate sanitization are extended from the work in Robinette et al. (2023): the mean squared error (MSE) and peak-signal-to-noise ratio (PSNR), which measure the absolute error between corresponding pixels of the reference and the altered images. In addition to these metrics, we also utilize the structural similarity index measure (SSIM), which measures the perceptual quality of an image or video in comparison to a reference. We utilize these metric implementations from the *scikit-image* metrics library. For more information on these metrics, please see the supplementary material.

## 3 Diffusion Model Sanitization

Our goal is to improve the image quality of sanitized images using a diffusion model approach to sanitization. A diffusion model is a generative model that consists of two main features: a forward diffusion process and a reverse diffusion process. In the forward diffusion process, an input image $x_0$ from a given data distribution $x_0 \sim q(x_0)$ is perturbed at timestep $t$ with Gaussian noise with a variance $\beta_t \in (0, 1)$ to produce a sequence of latent images $x_0, x_1, ..., x_T$. This forward noising process is defined by equations 1 and 2 and is commonly reparameterized by equation 3, which allows for direct sampling of noised latents at arbitrary steps. Here, $\alpha_t := 1 - \beta_t$, $\bar{\alpha}_t := \prod_{s=0}^{t} \alpha_s$, and $1 - \bar{\alpha}_t$ represents the variance of the noise for an arbitrary timestep.

$$q(x_1, \ldots, x_T | x_0) := \prod_{t=1}^{T} q(x_t | x_{t-1}) \tag{1}$$

$$q(x_t | x_{t-1}) := \mathcal{N}\left(x_t; \sqrt{1 - \beta_t} x_{t-1}, \beta_t \mathbf{I}\right) \tag{2}$$

$$q(x_t | x_0) = \mathcal{N}\left(x_t; \sqrt{\bar{\alpha}_t} x_0, (1 - \bar{\alpha}_t) \mathbf{I}\right) \tag{3}$$

---

[3]**Invoke-PSImage:** https://github.com/peewpw/Invoke-PSImage

---

**Algorithm 1** Hide, Sanitize, and Evaluate

---

1: **procedure** SANITIZEANDEVALUATE $(C, \mathcal{P}, \mathcal{H}, \mathcal{R})$
2:     $S \leftarrow \text{RandomPermute}(C)$                                      ▷ Shuffle covers to get secrets
3:     $C' \leftarrow \mathcal{H}(C, S)$ where $\mathcal{H} \in \{\mathcal{H}_{lsb}, \mathcal{H}_{ddh}, \mathcal{H}_{udh}\}$             ▷ Make containers
4:     $S' \leftarrow \mathcal{R}(C')$ where $\mathcal{R} \in \{\mathcal{R}_{lsb}, \mathcal{R}_{ddh}, \mathcal{R}_{udh}\}$     ▷ Reveal secrets prior to sanitization
5:     $\hat{C} \leftarrow \mathcal{P}(C')$ where $\mathcal{P} \in \{\text{DM-SUDS}, \text{SUDS}\}$             ▷ Sanitize containers
6:     $\hat{S} \leftarrow \mathcal{R}(\hat{C})$ where $\mathcal{R} \in \{\mathcal{R}_{lsb}, \mathcal{R}_{ddh}, \mathcal{R}_{udh}\}$     ▷ Reveal secrets post sanitization
7:     $\text{mse}_1, \text{psnr}_1, \text{ssim}_1 \leftarrow \text{CalcImageMetrics}(C', \hat{C})$         ▷ Image Preservation
8:     $\text{mse}_2, \text{psnr}_2, \text{ssim}_2 \leftarrow \text{CalcImageMetrics}(S, \hat{S})$         ▷ Secret Elimination
9:     **return** $\text{mse}_1, \text{psnr}_1, \text{ssim}_1, \text{mse}_2, \text{psnr}_2, \text{ssim}_2$           ▷ Return metrics
10: **end procedure**

---

The model then learns to reverse this diffusion process by refining the noised sample until it resembles a sample from the target distribution. The posterior $q(x_{t-1}|x_t, x_0)$ can be calculated using Bayes theorem in terms of $\tilde{\beta}_t$ and $\tilde{\mu}_t(x_t, x_0)$, which are defined in the equations below:

$$\tilde{\beta}_t := \frac{1 - \bar{\alpha}_{t-1}}{1 - \bar{\alpha}_t} \beta_t \tag{4}$$

$$\tilde{\mu}_t(x_t, x_0) := \frac{\sqrt{\bar{\alpha}_{t-1}}\beta_t}{1 - \bar{\alpha}_t} x_0 + \frac{\sqrt{\alpha_t(1 - \bar{\alpha}_{t-1})}}{1 - \bar{\alpha}_t} x_t \tag{5}$$

$$q(x_{t-1}|x_t, x_0) = N\left(x_{t-1}; \tilde{\mu}(x_t, x_0), \tilde{\beta}_t \mathbf{I}\right) \tag{6}$$

To represent $\mu_\theta(x_t, t)$ for the reverse diffusion process, we use a U-Net model to predict the noise $\epsilon$ added to the input image. The original image can then be predicted using equation 7.

$$x_0 = \frac{1}{\sqrt{\alpha_t}}\left(x_t - \frac{\beta_t}{\sqrt{1 - \bar{\alpha}_t}}\epsilon\right) \tag{7}$$

While diffusion models are most commonly used for generative purposes, we instead make use of the denoising capabilities developed during training. To use the diffusion model as a sanitizer, we can apply $t$ timesteps of noise to a potential container using equation 3 with a cosine beta scheduler from Ho et al. (2020), predict the noise using a neural network, and then refine the image using equation 7, effectively preserving the image quality while maintaining sanitization performance. As this is a blind approach to sanitization, prior knowledge of steganography techniques is not required for the training process. To highlight this feature of the framework and to emphasize the accessibility of this approach, we make use of a state-of-the-art, publicly available pretrained diffusion model from Nichol & Dhariwal (2021)[4]. For the remainder of this paper, the diffusion model sanitization process is referred to as DM-SUDS. An image sanitized with this process is denoted as $\hat{C}_{\text{DM}}$, and an attempted revealed secret from $\hat{C}_{\text{DM}}$ is denoted as $\hat{S}$.

## 4 EXPERIMENTS

In this section, we evaluate the sanitization capabilities of DM-SUDS compared to SUDS (see section 2.2), which is the only other blind deep learning sanitization approach tested against image secrets.

### 4.1 SETUP

To compare DM-SUDS and SUDS, we evaluate their sanitization performance on image secrets hidden with LSB, DDH, and UDH steganography using the CIFAR-10 test dataset (10000 RGB images). This process is demonstrated in algorithm 1. The CIFAR-10 dataset was chosen as it was used to demonstrate SUDS ability to sanitize RGB images, and it is also where the reconstruction

---

[4]Diffusion Model: https://github.com/openai/improved-diffusion

capabilities of SUDS start to break down. As this is the issue we are trying to address with DM-SUDS, it provides a great comparison playground for the two approaches. In order to provide fair comparisons, the same cover and secret combinations used to make containers are kept constant for each hiding technique, and DM-SUDS and SUDS are also evaluated on the same containers. The SUDS model used during this evaluation is from the repository provided by the authors. For the DM-SUDS model discussed in section 3, we use a timestep variable $t = 250$ in the forward and reverse diffusion processes. The image metrics discussed in section 2.3 (MSE, PSNR, and SSIM) are used to compare *covers $C$* with *sanitized images $\hat{C}$* (**Image Preservation**) as well as *secrets $S$* to *revealed secrets post sanitization $\hat{S}$* (**Secret Elimination**). To provide a baseline comparison to the effect that steganography has on images, a **None** column is also calculated, which uses these same metrics to compare *covers $C$* with *containers $C'$* (**Image Preservation**) as well as *secrets $S$* to *revealed secrets **pre** sanitization $S'$* (**Secret Elimination**).

## 4.2 RESULTS

**Sanitization**    From the results shown in table 1, DM-SUDS and SUDS are both able to successfully sanitize secrets as indicated by the high MSE values for **Secret Elimination**. In addition to the high MSE values, the low SSIM values also indicate that information retaining to the original secrets is eliminated as the similarity between the attempted reveal secret and the actual secret is small. Figure 2 demonstrates this successful sanitization as indicated by the last two columns of each subfigure showing the revealed secret before and after sanitization. As the true secret is not discernible from any of the samples hidden with a) LSB, b) DDH, or c) UDH steganography, this is a successful sanitization.

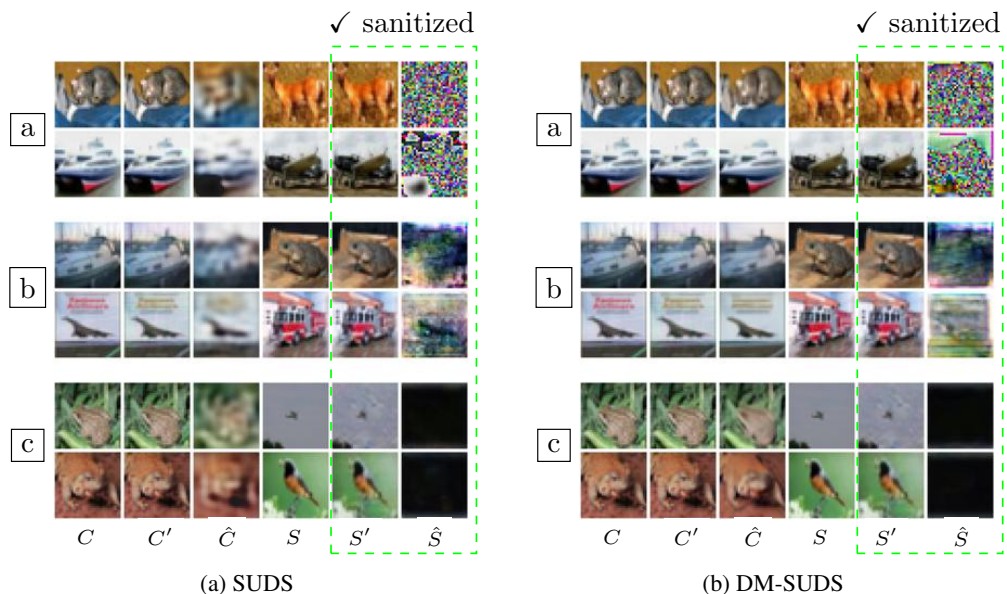

Figure 2: A comparison between DM-SUDS and SUDS sanitization for a) LSB, b) DDH, and c) UDH steganography. Sanitization performance is determined from the last two columns in each subfigure. As $S$ is not discernible from $\hat{S}$ for any of the approaches, sanitization is successful for both methods.

**Image Preservation**    While both methods successfully sanitize secrets, they differ largely in their ability to preserve image quality in the sanitization process. In the **Image Preservation** section of table 1, the MSE of the sanitized image compared to the original cover is smaller for DM-SUDS than SUDS for all steganography methods, achieving an average % improvement of 71.32% across LSB, DDH, and UDH. Additionally, the PSNR values are higher for DM-SUDS for all steganography methods as well. This indicates that sanitized images with DM-SUDS have a higher image quality with an average improvement of 22.43% for this metric. Lastly, the higher SSIM values of DM-SUDS demonstrate that the sanitized images are even more structurally similar to the original covers

Table 1: SUDS vs. Diffusion Model Test Stats

| Compare | $\mathcal{H}$ | Metric | None | SUDS | DM-SUDS | % Imp. |
|---------|------|--------|------|------|---------|--------|
| Image Preservation | LSB | MSE | 37.69 | 382.33 | **103.52** | 72.92 |
| | | PSNR | 32.47 | 22.75 | **28.15** | 23.73 |
| | | SSIM | 0.96 | 0.76 | **0.91** | 19.74 |
| | DDH | MSE | 119.72 | 288.63 | **90.18** | 68.76 |
| | | PSNR | 28.46 | 23.93 | **28.74** | 20.10 |
| | | SSIM | 0.94 | 0.80 | **0.91** | 13.75 |
| | UDH | MSE | 23.91 | 392.03 | **108.65** | 72.29 |
| | | PSNR | 34.58 | 22.62 | **27.93** | 23.47 |
| | | SSIM | 0.97 | 0.76 | **0.90** | 18.42 |
| Secret Elimination | LSB | MSE | 79.29 | 9519.54 | 9147.77 | |
| | | PSNR | 29.17 | 8.46 | 8.65 | |
| | | SSIM | 0.97 | 0.01 | 0.02 | |
| | DDH | MSE | 86.91 | 5231.90 | 3300.94 | |
| | | PSNR | 29.54 | 11.32 | 13.29 | |
| | | SSIM | 0.95 | 0.08 | 0.21 | |
| | UDH | MSE | 157.05 | 14985.76 | 15201.37 | |
| | | PSNR | 26.75 | 6.92 | 6.84 | |
| | | SSIM | 0.91 | 0.03 | 0.03 | |

compared to using SUDS (average 17.30% improvement). All three metric values demonstrate that DM-SUDS has better image preservation capabilities compared to sanitization with SUDS. Figure 3 visually compares an original cover to images sanitized with Gaussian noise, SUDS, and DM-SUDS. While some fine details are not reproduced in the sanitized images via DM-SUDS (e.g., muscles along the neck of the horse), the image quality compared to Gaussian noise and SUDS is drastically better. Gaussian noise causes significant damage to the image, and the blurry nature of the reconstructed SUDS image compromises both its visual appeal and informational content. DM-SUDS, therefore, improves sanitization image quality while maintaining sanitization abilities, outperforming SUDS by 71.32% (MSE), 22.43% (PSNR), and 17.30% (SSIM).

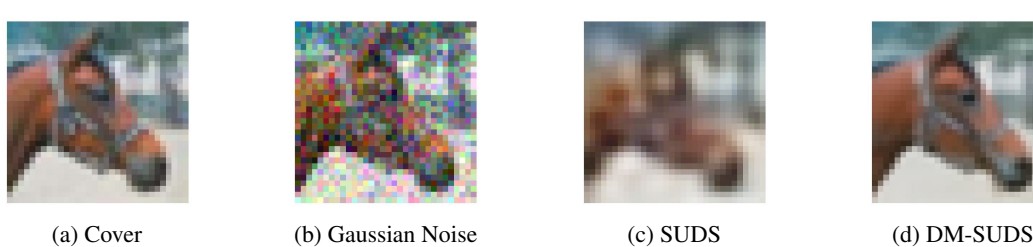

| (a) Cover | (b) Gaussian Noise | (c) SUDS | (d) DM-SUDS |

Figure 3: Image preservation comparison between an original cover and sanitization with Gaussian noise, SUDS, and DM-SUDS. DM-SUDS achieves the highest image quality of a sanitized image.

## 5 ANALYSIS AND DISCUSSION

**Diffusion Steps** For the previous experiments, we utilized a timestep $t = 250$ in the diffusion process, which means that we sample a noisy version of the input image at $t = 250$. In the reverse diffusion process, this number is then encoded, and the neural network estimates the added noise to the image from this timestep. To evaluate the robustness of the diffusion model approach and the pervasiveness of the selected steganographic hiding methods, we evaluate the diffusion model at various timesteps $t$ as shown in figure 4. In regard to the sanitized image $\hat{C}$, with too many timesteps, the image quality of the reconstructed image starts to deteriorate, as shown by $t = 1000$. The blurry nature of the resulting processed image indicates that the more noise that is added to a container, the more difficult it is to predict the amount of added noise, affecting the resulting refined image $\hat{C}$. With too few timesteps, however, the secret persists after sanitization (see DDH $\hat{S}$). This

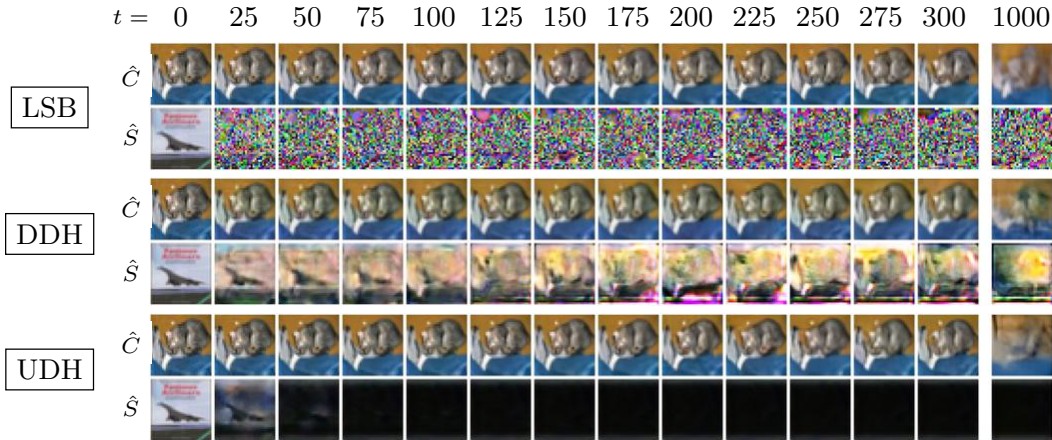

Figure 4: DM-SUDS with varying forward diffusion timesteps. With too few timesteps, secrets are not sanitized, and with too many timesteps, the refined image quality degrades.

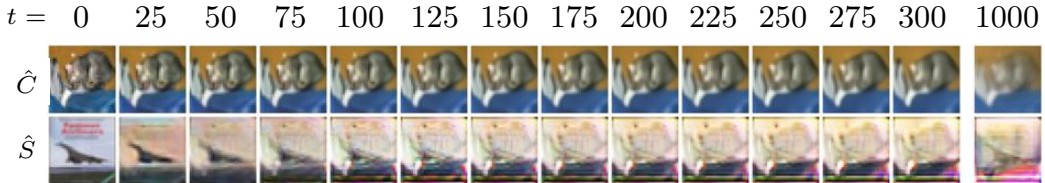

Figure 5: Sanitized images for DDH steganography with no added noise in the diffusion process (skip forward diffusion). Even at $t = 1000$ timesteps, the secret survives, meaning that added noise is necessary for the sanitization process.

is particularly important for secrets hidden with DDH, which seem to persist up until $t = 125$. This is also an interesting result for the robustness of each hiding method. The order of robustness as determined by the persistence of the secret with increased timesteps is DDH, UDH, and LSB. DDH, therefore, is a more robust hiding method than the other two. While the number of selected timesteps impacts performance near the edges, this value is not particularly sensitive.

**Direct Denoising** In the majority of this work, we sample a noisy image of the container at timestep $t$, and then denoise this image to a sanitized version of the input. Another way to sanitize via DM-SUDS is to treat the potentially steganographic image as the direct input to the denoiser, skipping the forward diffusion process. With this approach, however, we still have to provide an estimate $t$ as input to the reverse diffusion neural network. To evaluate this approach, we sanitize directly from the containers hidden with DDH at various timesteps, as DDH creates the most persistent secrets (see the previous paragraph). From the results shown in figure 5, the secret is never really sanitized from DDH as indicated by the $t = 1000$ column. Even at this timestep, the secret is still distinguishable. This indicates that the true power of the diffusion model sanitization approach lies in the added Gaussian noise in the forward diffusion process.

**Evaluation on ImageNet** As the reconstruction ability of SUDS decreases with an increase in image complexity Robinette et al. (2023), we seek to determine if DM-SUDS is able to maintain its abilities with more complex images using the ImageNet dataset with images of size $\mathcal{R}^{3 \times 64 \times 64}$. For this experiment, we evaluate the sanitization of ImageNet secrets hidden with DDH. DDH was chosen as it is the most robust of the tested hiding methods as determined by the experiments conducted in *Diffusion Steps*. From the results shown in figure 6, DM-SUDS also works for more complex datasets, as the image preservation is conserved ($C$ to $\hat{C}$), and the secrets are successfully sanitized ($S$ to $\hat{S}$). This result is further verified by the metrics presented in table 2.

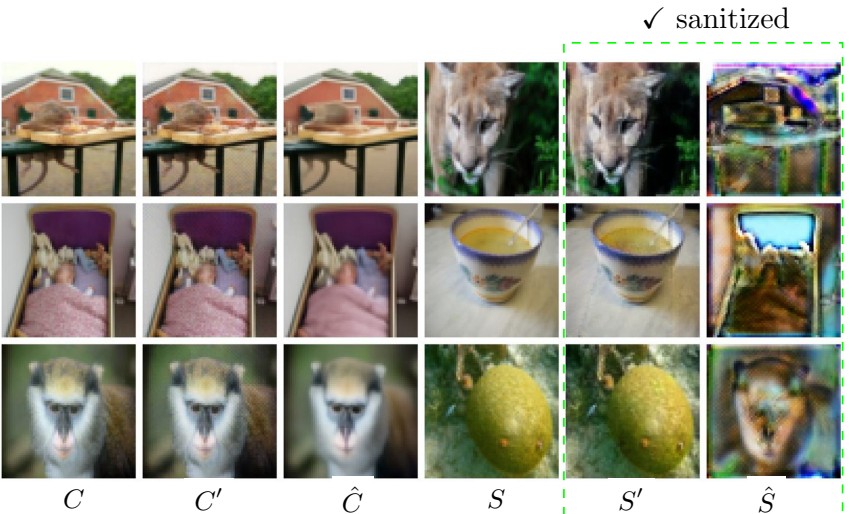

Figure 6: DM-SUDS sanitization on ImageNet steganography using DDH. Secrets are successfully sanitized, as indicated by the last two columns – the secret is not discernable from $\hat{S}$.

Table 2: DM-SUDS ImageNet Image Metrics

| Compare | Metric | DM-SUDS |
|---|---|---|
| Image Preservation | MSE | 68.57 |
| | PSNR | 30.21 |
| | SSIM | 0.88 |
| Secret Elimination | MSE | 7664.38 |
| | PSNR | 9.60 |
| | SSIM | 0.07 |

**Potential Improvements and Use Cases** While DM-SUDS successfully outperforms SUDS in the sanitization of steganography, there are areas for future improvement. One potential drawback of this approach (as with SUDS) is that DM-SUDS is trained on a specific data distribution. While it may successfully sanitize images of various distributions, the reconstructed image quality will be sacrificed. Most use cases, however, will likely be of the same distribution. For instance, a bank company's network traffic might contain images of customer identification documents, signature cards, check images, etc., which would exhibit the same data distribution. DM-SUDS could be used to protect the exfil of proprietary information of that bank from bad actors, as well as bad actors attempting to introduce malware payloads into the system via steganography.

## 6 CONCLUSION

In this work, we introduce a novel blind deep learning steganography sanitization method that utilizes a diffusion model framework to restore images called DM-SUDS. We demonstrate the success of such an approach compared to SUDS, a previous VAE-based approach, and analyze features specific to a diffusion model to wholistically evaluate DM-SUDS. Where SUDS has difficulty in reconstructing images, DM-SUDS achieves a higher reconstruction ability while maintaining sanitization performance, improving image preservation MSE by 71.32%, PSNR by 22.43% and SSIM by 17.3%. One of the most impactful features of DM-SUDS lies in its accessibility, as any pretrained diffusion model of a target domain's data distribution can be implemented to protect a system against steganography. This work, therefore, introduces a highly effective and beneficial use case for a diffusion model while marking a significant advancement in the field of steganography sanitization. In the future, we hope to focus on optimizing DM-SUDS further and explore its application in various domains.

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
