# Supplementary Material to: Monsters in the Dark: Sanitizing Hidden Threats with Diffusion Models

In this supplementary material, dive deeper into the steganographic techniques used in this work and present additional sanitization results.

## 1 Steganography Details

### 1.1 Least Significant Bit Method

Digital images are a commonly used cover medium, which are composed of pixels, or a finite set of digital values in a two-dimensional array. While there are various models for representing colors in pixels, the RGB color model utilizes red, green, and blue color channels. Each pixel of an image is then composed of three 8-bit values which represent the intensity of each color, as shown in Figure 1. Here, the RGB value of the indicated pixel is RGB(160, 92, 100) or represented in binary RGB(10100000, 01011100, 01100100). In binary, the leftmost bit of the binary digit is known as the most significant bit, and the rightmost bit is known as the least significant bit. This results from the degree of change that can occur in the 8-bit binary digit value if this bit is switched. For example, if the leftmost bit of 11111111 (or 255) is flipped, the number becomes 01111111 (or 127), changing by approximately 50%. If, however, the rightmost bit is changed from 11111111 (255) to 11111110 (254), the value only changes by approximately 0.4%. In this way, information can be embedded in the least significant bits of an RGB value while imperceptibly changing the visual rendering of the image. Figure 2 shows the effects of flipping the two least significant bits of every color channel in the pixels of the original image. The resulting difference between the two images are imperceptible to the Human Visual System (HVS).

For images of the same size, it is common to hide the four most significant bits of the secret image in the four least significant bits of the cover image, which was first introduced in Kurak & McHugh (1992). As such, this work makes use of the four least significant bits to hide secrets via the LSB implementation.

### 1.2 Dependent Deep Hiding

Dependent deep hiding (DDH) differs from a traditional hiding method in that it utilizes a deep learning model for the hide and reveal functions, as shown in figure 3a. The hide network takes as

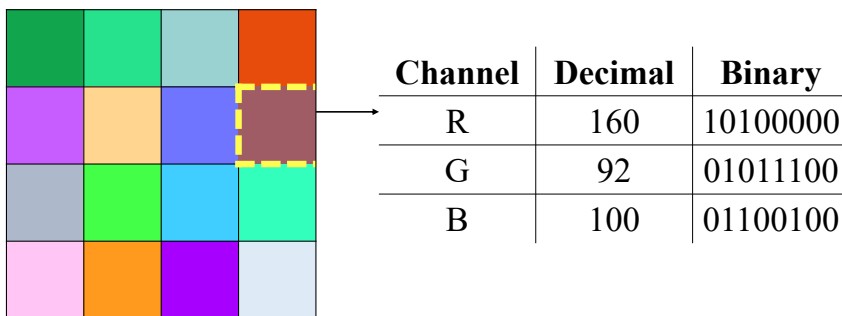

Figure 1: Image autopsy demonstrating the red, green, and blue color channels of a single pixel in an image.

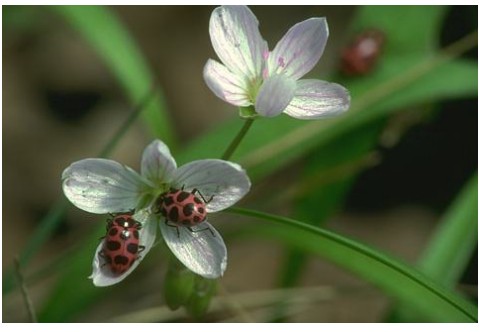 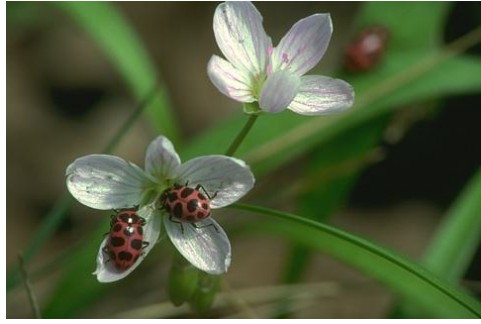

(a) Original Image                (b) Altered Image

Figure 2: A demonstration of the LSB method for hiding images. The original image 2a appears identical to be the altered image 2b. The altered image shows the effects of flipping the two least significant bits of each channel (red, green, and blue) of every pixel in the image.

input a cover and a secret and produces a container image. The reveal network then takes as input the produced container and maps this image to a revealed secret. The hide and reveal models are usually trained in tandem, and the loss function minimizes the difference between the container and cover, as well as the revealed secret and original secret. Recent CNN-based and GAN-based methods include SGAN Volkhonskiy et al. (2020), UT-GAN Yang et al. (2019), ASDL-GAN Tang et al. (2017), SPAR-RL Tang et al. (2020), R-GAN Wu et al. (2020), HIDDeN Zhu et al. (2018), SSteGAN Wang et al. (2018), and Deep Steganography Baluja (2017). This work utilizes a CNN-based implementation available from a code base that also contains a universal deep hiding implementation[1]. This code base was chosen because of its dual (UDH and DDH) functionality. By utilizing implementations from the same code base, we are better able to accurately compare the methodologies (DDH vs. UDH) as differences in programming implementations can affect training performance Henderson et al. (2018).

---

[1]https://github.com/ChaoningZhang/Universal-Deep-Hiding

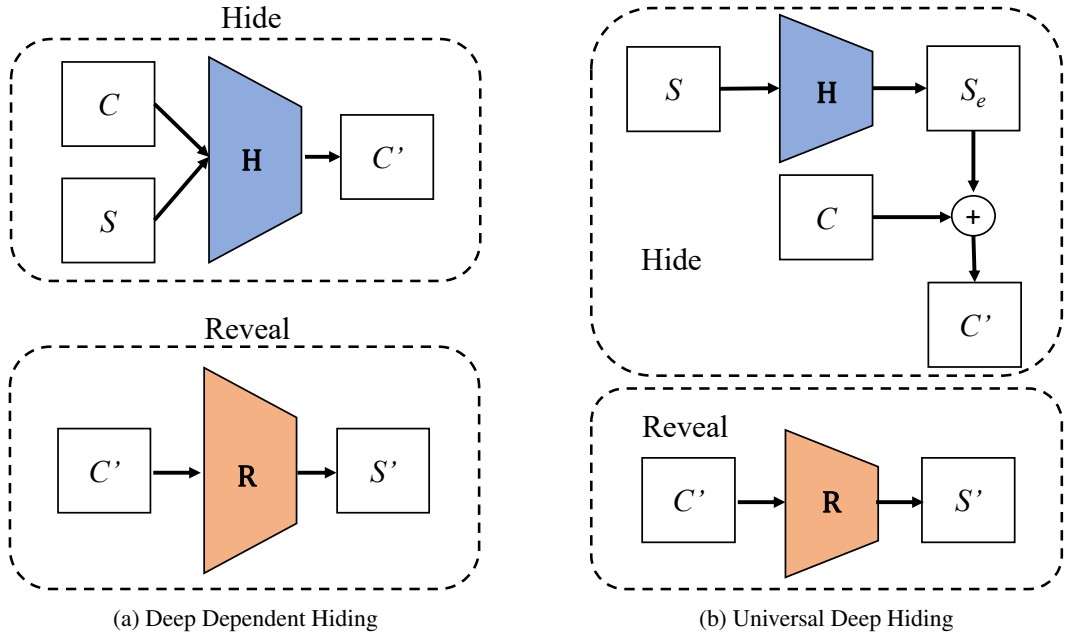

(a) Deep Dependent Hiding

(b) Universal Deep Hiding

Figure 3: A comparison between dependent deep hiding (DDH) and universal deep hiding (UDH). In 3a, a cover $C$ and secret $S$ are used as inputs to the hide network $\mathcal{H}$, producing a container image $C'$. The reveal network of DDH takes a container as an input and maps this image to the revealed secret $S'$. While DDH is cover dependent, UDH only relies on the secret. The secret, therefore, acts as the only input into the hide network $\mathcal{H}$ in 3b, which produces a secret noise image $S_e$. $S_e$ can then be added to any arbitrary cover image to produce a container. The reveal network $\mathcal{R}$ of UDH maps this container to a revealed secret, which resembles the original secret.

## 1.3 UNIVERSAL DEEP HIDING

Universal deep hiding (UDH) is similar to DDH in that it also utilizes deep learning, but UDH is cover independent Zhang et al. (2020). In figure 3b, the input to the hide network is just the secret, which produces a corresponding secret noise image $S_e$. $S_e$ is then added with an arbitrary cover to produce a container image. A reveal network is then used to reproduce the original secret. The secret noise image can be applied to any cover, and the secret is still retrievable by the reveal network. The hide and reveal networks are trained in tandem in UDH as well.

## 2 IMAGE METRICS

The equations for each of these metrics (MSE, PSNR, SSIM) are shown in equations 1, 2, and 3, where $A$ and $B$ are the compared images of size (c, h, w), $MAX$ is the maximum possible pixel value (for a given bit depth), $\mu_A$ and $\mu_B$ are the average values of images A and B, $\sigma_A^2$ and $\sigma_B^2$ are the variances of images A and B, $\sigma_{AB}$ is the covariance of A and B, and $c_1$ and $c_2$ are constants to avoid division by zero.

$$MSE(A, B) = \frac{1}{chw} \sum_{i=1}^{c} \sum_{j=1}^{h} \sum_{k=1}^{w} (A_{i,j,k} - B_{i,j,k})^2 \qquad (1)$$

$$PSNR(A, B) = 10 \log_{10} \left( \frac{MAX^2}{MSE(A, B)} \right) \qquad (2)$$

$$SSIM(A, B) = \frac{(2\mu_A\mu_B + c_1)(2\sigma_{AB} + c_2)}{(\mu_A^2 + \mu_B^2 + c_1)(\sigma_A^2 + \sigma_B^2 + c_2)} \qquad (3)$$

## 3 SANITIZATION

The following section includes larger images of the images included in the main body of the paper (fig. 2) or more images of the same analysis.

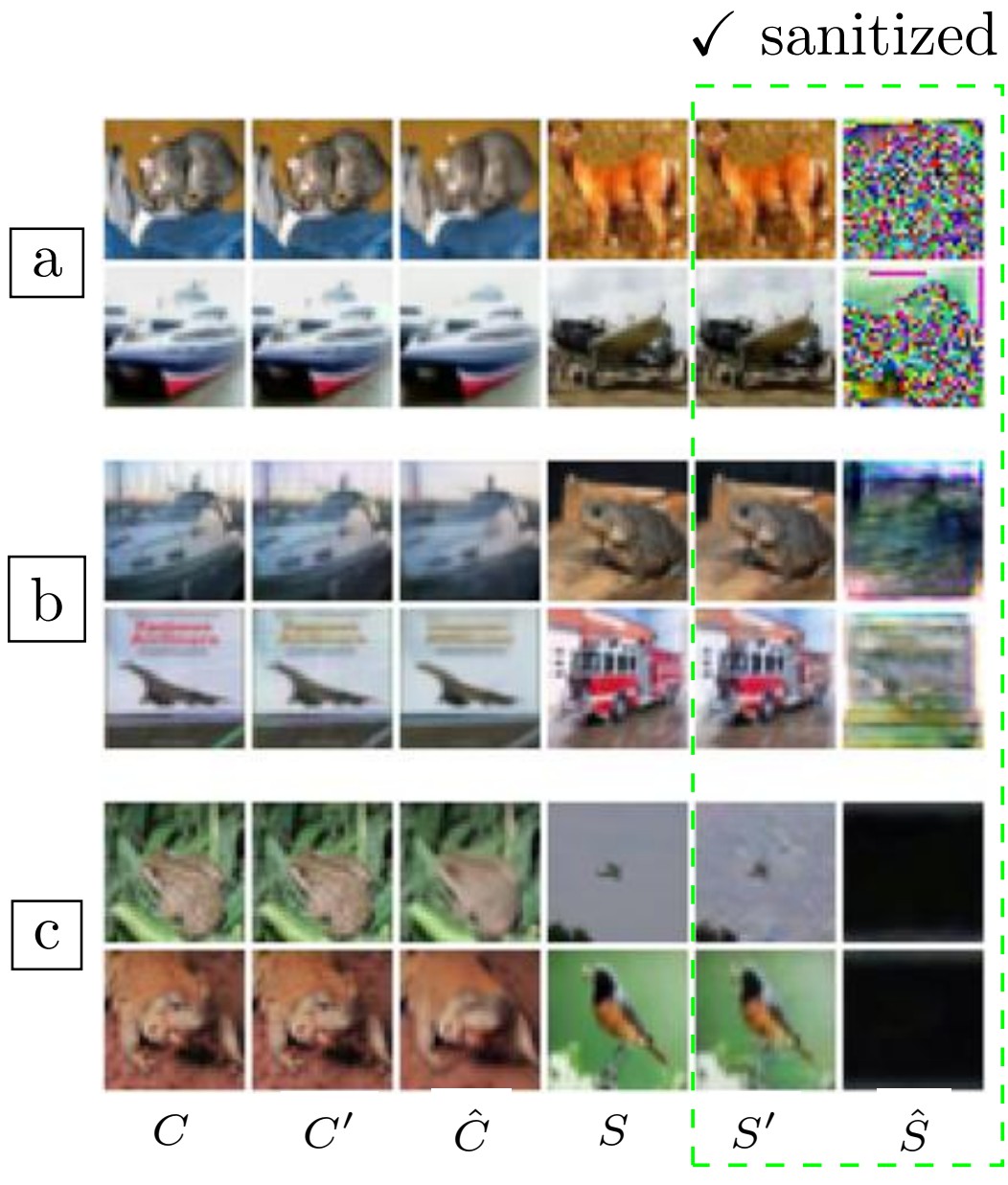

Figure 4: DM-SUDS sanitization where a) are containers hidden with LSB, b) DDH, and c) UDH.

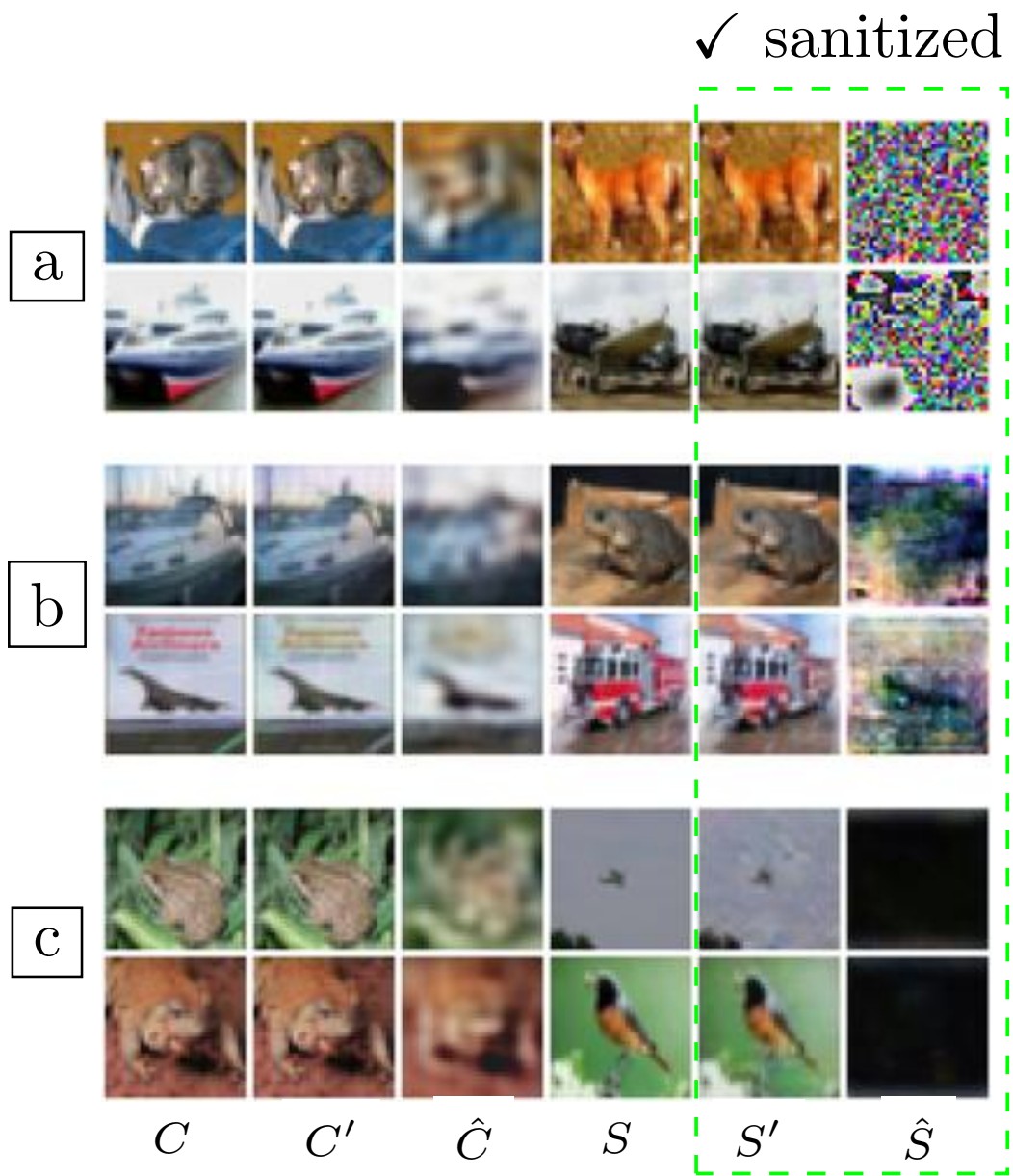

$$C \qquad C' \qquad \hat{C} \qquad S \qquad S' \qquad \hat{S}$$

Figure 5: SUDS sanitization where a) are containers hidden with LSB, b) DDH, and c) UDH.

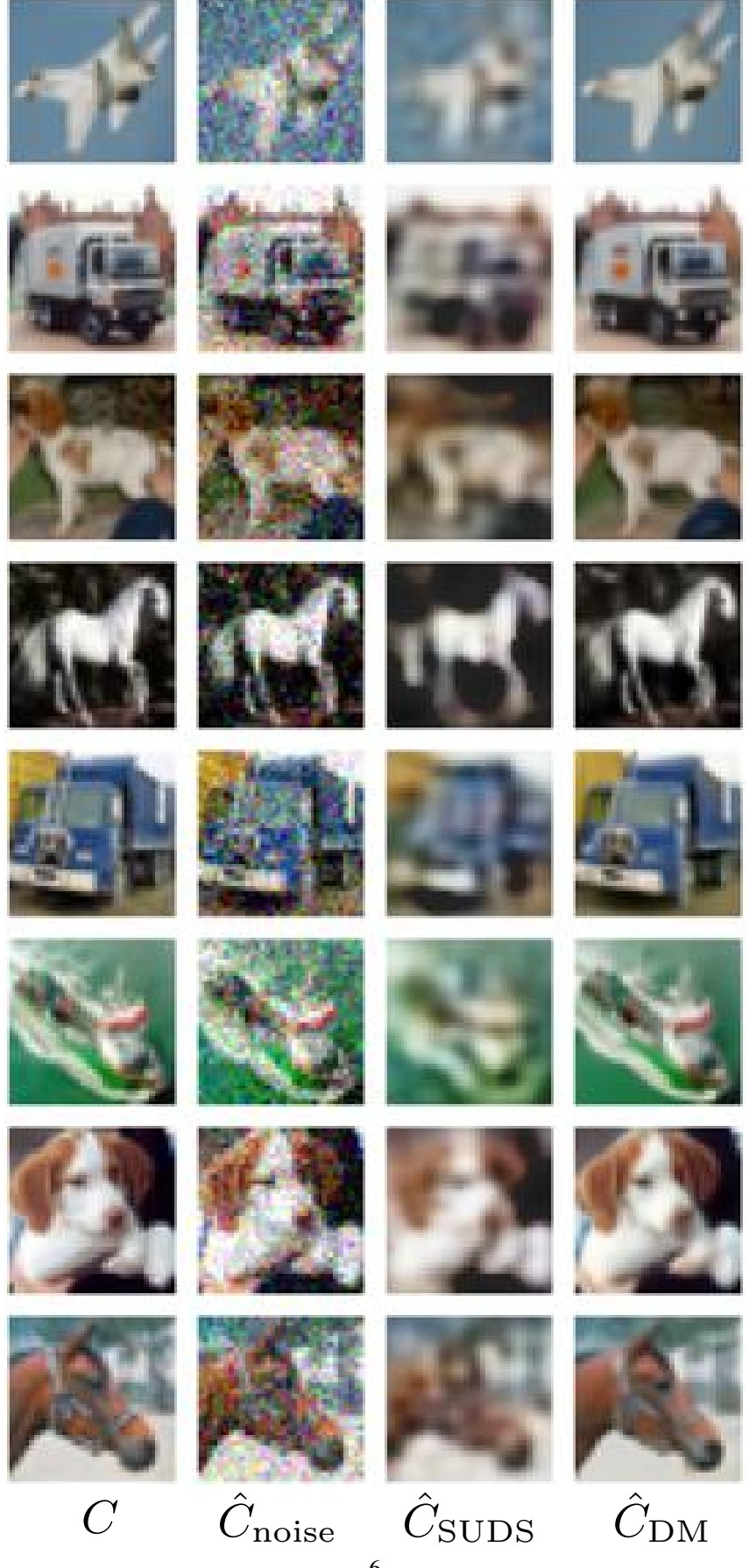

$$C \qquad \hat{C}_{\text{noise}} \qquad \hat{C}_{\text{SUDS}} \qquad \hat{C}_{\text{DM}}$$

Figure 6: Comparison among the image preservation capabilities of Gaussian noise, SUDS, and DM-SUDS sanitization techniques.

✓ sanitized

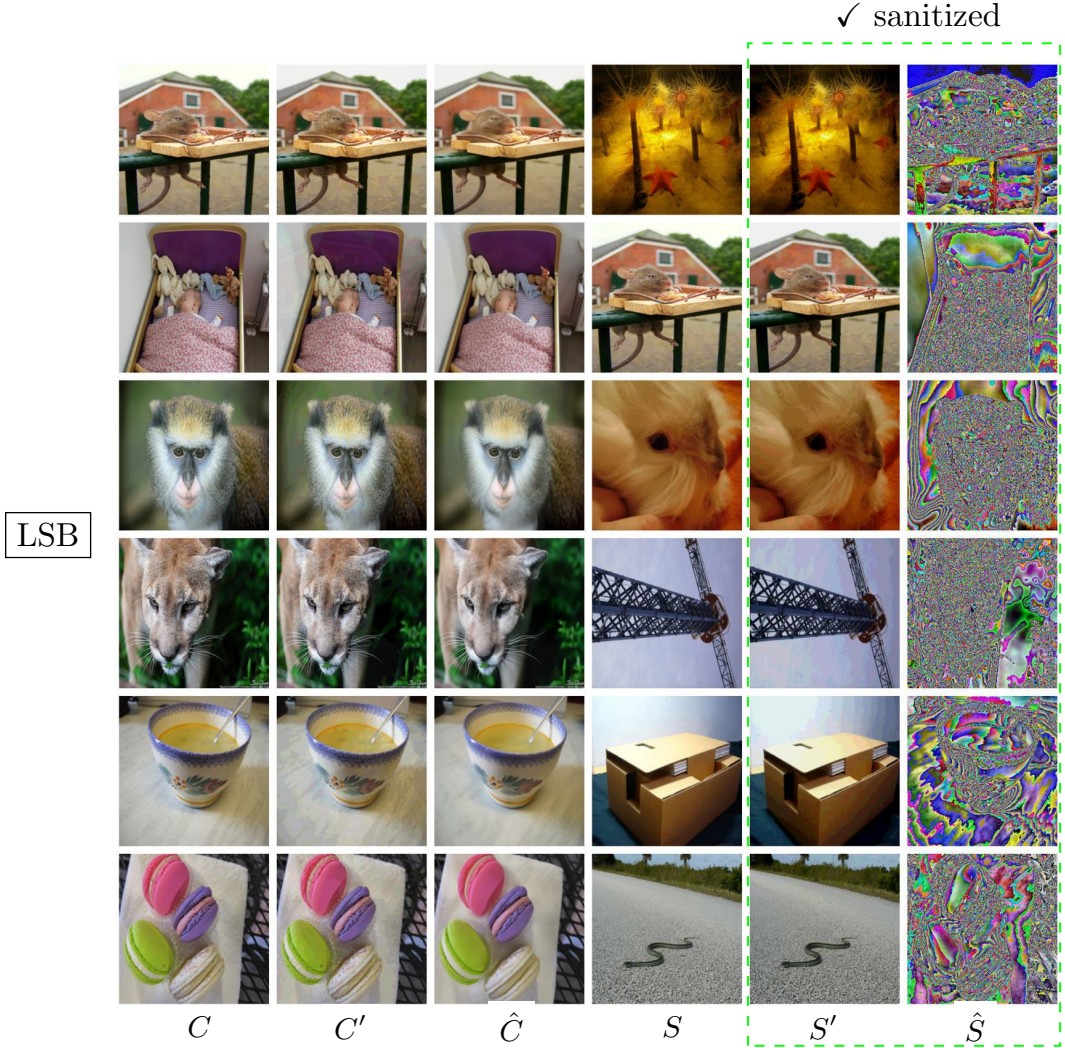

$C$        $C'$        $\hat{C}$        $S$        $S'$        $\hat{S}$

Figure 7: DM-SUDS capabilities on the ImageNet dataset.