# OpenReview forum: "Monsters in the Dark: Sanitizing Hidden Threats with Diffusion Models"
_ICLR.cc/2024/Conference — Submitted to ICLR 2024_

### Official Review · Reviewer_sHuH · 2023-10-28

**Soundness:** 3 good
**Presentation:** 3 good
**Contribution:** 3 good
**Rating:** 6
**Confidence:** 3

**Summary:**

This paper proposes a novel image sanitization approach based on diffusion models. The method achieves better image quality preservation compared with the state-of-the-art SUDS (a paper published in 2023).

**Strengths:**

+ The contribution of this work is well-positioned.
+ The experiments support the contribution well.

**Weaknesses:**

- It would good if the authors can delve into the mechanism of why diffusion model can help improve the image quality preservation of SUDS. For example, provide more discussions.
- The methodology of this work is somehow incremental. If the authors could address more clearly about the advantages of introducing the diffusion (even with some demonstrative experiments), it would be better.

**Questions:**

As mentioned in the weakness part.

---

> ### Author Response · Authors · 2023-11-21
> **Reviewer Feedback and Questions**
>
> Thank you for your review. We have addressed as many of your concerns as possible. Please see our responses below. (R) refers to you, the reviewer, and (A) indicates our response.
>
> (R) *It would good if the authors can delve into the mechanism of why diffusion model can help improve the image quality preservation of SUDS. For example, provide more discussions*
>
> (A) Thank you for your comment. We attempt to address this in Section 5, *Direct Denoising*. We find that the true power of the sanitization process occurs in the forward denoising process. We believe that the image quality preservation of DM-SUDS beats SUDS because of the choice of network: U-Net compared to a VAE. Where information is lost in a VAE architecture, the skip connections of a U-Net allow more relevant information to persist after the bottleneck.
>
> (R) *The methodology of this work is somehow incremental. If the authors could address more clearly about the advantages of introducing the diffusion (even with some demonstrative experiments), it would be better.*
>
> (A) Thank you for this suggestion. We believe the benefits of using DM-SUDS really shine in Section 4.2. Here, we see that DM-SUDS is better able to preserve image quality while maintaining sanitization performance.

---

### Official Review · Reviewer_158U · 2023-10-28

**Soundness:** 3 good
**Presentation:** 3 good
**Contribution:** 2 fair
**Rating:** 5
**Confidence:** 5

**Summary:**

This manuscript first proposed a diffusion model framework to sanitize steganography and preserve image quality in the sanitization process. However, the literature research on the field of sanitation hidden message is insufficient, and there is a lack of experiments, not only the sanitation evaluations on various robust steganography [1~5], but also the comparison with other sanitation methods [6~10].

**Strengths:**

(1) This manuscript first utilizes a diffusion model framework to sanitize universal and dependent steganography.

(2) This manuscript is well written.

**Weaknesses:**

（1）	The application scenario in this article is not clearly described. The two use cases mentioned in the penultimate paragraph of the manuscript have been replaced by robust steganography, a more suitable tool. Robust steganography has been developed for many years [1~5] and is not mentioned in this article, which is a lack of research.

（2）	This manuscript lacks research on relevant literature of sanitization methods for robust information hiding [6~10]. The authors should not define their previous work [11] as the state of the art easily.

（3）	At the end of the abstract, the authors propose DM-SUDS is the first blind deep learning image sanitization framework to meet these image quality results. I suggest the authors read the literature [10].

（4）	In subsection of 2.2 SANITIZATION, it is not rigorous to indicate that text-based secrets are more fragile than image-based secrets. The authors should conduct experiments to verify the robustness of text-based secrets embedded by robust steganography (such as DMAS [1]) and image-based secrets hidden by UDH [12].

（5）	In section of ANALYSIS AND DISCUSSION, the DM-SUDS also be evaluated on ImageNet. Conducting more experiments with more complex images is indeed necessary, however, the evaluation is limited only to LSB. We all know that LSB is not robust and can be disabled by common JPEG compression. It is meaningless to sanitize the secret messages embedded by LSB. LSB method can not transmit messages over lossy channels such as online social networks.



Reference:

[1] Zhang, Yi, et al. "Dither modulation based adaptive steganography resisting JPEG compression and statistic detection." Multimedia Tools and Applications 77 (2018): 17913-17935.

[2] Zhao, Zengzhen, et al. "Improving the robustness of adaptive steganographic algorithms based on transport channel matching." IEEE Transactions on Information Forensics and Security 14.7 (2018): 1843-1856.

[3] Zeng, Kai, et al. "Robust Steganography for High Quality Images." IEEE Transactions on Circuits and Systems for Video Technology (2023): 4893-4906.

[4] Zeng, Kai, et al. "Upward Robust Steganography Based on Overflow Alleviation." IEEE Transactions on Multimedia (2023).

[5] Lan, Yuhang, et al. "Robust image steganography: hiding messages in frequency coefficients." Proceedings of the AAAI Conference on Artificial Intelligence. Vol. 37. No. 12. 2023.

[6] Li, Qi, et al. "Concealed attack for robust watermarking based on generative model and perceptual loss." IEEE Transactions on Circuits and Systems for Video Technology 32.8 (2021): 5695-5706.

[7] Hatoum, Makram W., et al. "Using deep learning for image watermarking attack." Signal Processing: Image Communication 90 (2021): 116019.

[8] Zhu, Zhiying, et al. "Destroying robust steganography in online social networks." Information Sciences 581 (2021): 605-619.

[9] Wei, Ping, et al. "Breaking Robust Data Hiding in Online Social Networks." IEEE Signal Processing Letters 29 (2022): 2682-2686.

[10] Zhu, Zhiying, et al. "Image Sanitization in Online Social Networks: A General Framework for Breaking Robust Information Hiding." IEEE Transactions on Circuits and Systems for Video Technology (2022): 3017-3029.

[11] Robinette, Preston K., et al. "Suds: Sanitizing universal and dependent steganography." arXiv preprint arXiv:2309.13467 (2023).

[12] Zhang, Chaoning, et al. "Udh: Universal deep hiding for steganography, watermarking, and light field messaging." Advances in Neural Information Processing Systems 33 (2020): 10223-10234.

**Questions:**

See Weaknesses

---

> ### Author Response · Authors · 2023-11-21
> **Reviewer Feedback and Questions**
>
> We would first like to thank you for your response and time in sharing the above resources. We have addressed as many of your concerns as possible. Please see our responses below. (R) refers to you, the reviewer, and (A) indicates our response.
>
> (R) *"The application scenario in this article is not clearly described. The two use cases mentioned in the penultimate paragraph of the manuscript have been replaced by robust steganography, a more suitable tool. Robust steganography has been developed for many years [1~5] and is not mentioned in this article, which is a lack of research."*
>
> (A) We thank you for your feedback and appreciate you mentioning robust steganography. Robust steganography and corresponding techniques were not included in this work, as we assume in the paper that all steganographic techniques are meant to survive modifications to the container. Additionally, most robust steganographic techniques lie in the domain of traditional steganography and are used to embed text secrets or grayscale images, where we consider RGB images. We can therefore hide an image secret which is 3x larger (RGB: 3xhxw compared to Grayscale: 1xhxw).
>
> (R) *"This manuscript lacks research on relevant literature on sanitization methods for robust information hiding [6~10]. The authors should not define their previous work [11] as the state of the art easily."*
>
> (A) The authors would first like to thank you for your added literature. We have updated our literature review in the paper to include these works.  In [6], the authors create a watermarked image in the training process to learn a successful attack. [7] uses STDM and SS watermarked images from BOSS to train the Fully Convolutional Neural Network Denoising Attack (FCNNDA). This approach uses text-based secrets and is also non-blind. The method proposeds in [8-mentioned in paper] and [9] use containers created with DMAS (text secrets) in the training process. This means that they are not blind and are using a different type of secret. While [10] could possible be considered blind (wasn't able to find code and steg was used to train parameters), they only consider text secrets, which is also not directly applicable to our work. These are great additions to the literature review and help to better position our paper. Thank you for linking them in your review!
>
> (R) *"At the end of the abstract, the authors propose DM-SUDS is the first blind deep learning image sanitization framework to meet these image quality results. I suggest the authors read the literature [10]."*
>
> (A) Thank you for pointing this out. We are still unsure if [10] can be considered `blind` as the image quality of the approach is heavily dependent on the parameter tuning, which utilizes steganographic images. We do, however, cede that we can be more precise in our language and have updated this sentence to be: *DM-SUDS is the first blind deep learning image **secret** sanitization framework to meet these image quality results.* Thank you for highlighting this!
>
> (R) *In subsection of 2.2 SANITIZATION, it is not rigorous to indicate that text-based secrets are more fragile than image-based secrets. The authors should conduct experiments to verify the robustness of text-based secrets embedded by robust steganography (such as DMAS [1]) and image-based secrets hidden by UDH [12].*
>
> (A) Thank you for pointing this out. As we focus only on image-based secrets, we have removed this blanket statement from section 2.2. We do, however, believe that image-based secrets are more robust but agree that this statement should not be included without the experimentation to back it up. Thank you! We believe this will be an interesting area to explore in future work.
>
> (R) *"In section of ANALYSIS AND DISCUSSION, the DM-SUDS also be evaluated on ImageNet. Conducting more experiments with more complex images is indeed necessary, however, the evaluation is limited only to LSB. We all know that LSB is not robust and can be disabled by common JPEG compression. It is meaningless to sanitize the secret messages embedded by LSB. LSB method can not transmit messages over lossy channels such as online social networks."*
>
> (A) Thank you for this suggestion. We have updated these experiments in the paper to be conducted on DDH and believe this greatly strengthens our analysis. Please see section 5, *Evaluation on ImageNet*. We would also like to highlight that not all images are or can utilize JPEG compression. There are cases where lossless image types are required, such as medical imaging and forensic imaging for legal processes. Please see our reviewer response to Reviewer zZQW.

---

### Official Review · Reviewer_zZQW · 2023-10-28

**Soundness:** 2 fair
**Presentation:** 2 fair
**Contribution:** 1 poor
**Rating:** 3
**Confidence:** 5

**Summary:**

In my subjective opinion, authors are solving wrong problem. At the moment, steganography based on deep learning is very detectable (in the sense of evaluation under Kerkchoff's principle), since it generates images with non-natural noise. Moreover, the capacity of steganography based on deep learning is very inferior in terms of capacity to classical steganography by cover modification, where to my knowledge the state of the art is [1]. This means that rational attacker has a little incentive to use suboptimal methods, where better ones exists.

Needless to say, state of the art steganography is usually fragile, which means that changing a single pixel might (in modern scheme very likely will) make the extraction of the message impossible. With the respect, the proposed work is not solving the right problem, since modern algorithm certainly do not survive JPEG compression. Of course, there are works on making steganography robust [2], but at the expense of the capacity, as part of the capacity needs to be reserved for the error correction.

With respect to the above, I have found the work very shallow. It claims that existing techniques for steganography by cover modification does not work without even trying. Moreover, the comparison would not be made equal, because of differences in capacity, choice of the algorithm by the attacker.


[1] Bernard, Solène, et al. "Backpack: a Backpropagable Adversarial Embedding Scheme." IEEE Transactions on Information Forensics and Security 17 (2022): 3539-3554.

[2] Kin-Cleaves, Christy, and Andrew D. Ker. "Adaptive steganography in the noisy channel with dual-syndrome trellis codes." 2018 IEEE International Workshop on Information Forensics and Security (WIFS). IEEE, 2018.

[3] Solanki, Kaushal, Anindya Sarkar, and B. S. Manjunath. "YASS: Yet another steganographic scheme that resists blind steganalysis." Information Hiding: 9th International Workshop, IH 2007, Saint Malo, France, June 11-13, 2007, Revised Selected Papers 9. Springer Berlin Heidelberg, 2007.

**Strengths:**

I good list of prior art but misses some state of the art.

**Weaknesses:**

* I think the solved problem is not interesting.
* The experimental evaluation is poor. It misses prior art about which authors say it would not work (I would like to see it does not work). There is a lot of prior art in watermark removal. You should show that they do not work.
* The capacity of images of size 32x32 (size of images in Cifar 10) will be very small, which means that the experimental settings are distant from the reality

**Questions:**

* Have you tried basic JPEG compression and recompression with different quality factors?
* Why you have not tried methods from the famous stirmark test?
* What is the length of the message you have hidden into the images?

---

> ### Author Response · Authors · 2023-11-21
> **Reviewer Feedback**
>
> Thank you for your review. We have addressed as many of your concerns as possible. Please see our responses below. (R) refers to you, the reviewer, and (A) indicates our response.
>
> (R) *"In my subjective opinion, authors are solving wrong problem...This means that rational attacker has a little incentive to use suboptimal methods, where better ones exists."*
>
> (A)  The authors would first like to thank you for your honest feedback. In regard to steganography being considered `very detectable', we cannot argue against the performance of detection capabilities. We would like to highlight, though, that these detection models are trained on known steganographic methods. The training sets or statistical anomalies are collected from steganography techniques that already exist. Detection capabilities might then perform well at detecting containers made with these known techniques, but these help to safeguard the past. Steganography is a volatile field, especially since the application of deep neural networks to hide information introduced in [1]. This is definitely not an extensive list, but other works shortly after include [2-8]. As new steganographic techniques are being constantly introduced, how do we then protect the future as well? We believe that sanitization provides this assurance, as shown in the paper. DM-SUDS is able to protect against unknown steganography techniques (not used for training) while maintaining a high image quality. Please see Section 4.2 for this analysis. This is a very exciting result as it can protect against future techniques, not just known ones from the past.
>
> (R) *Needless to say, state of the art steganography is usually fragile .... as part of the capacity needs to be reserved for the error correction.*
>
> (A) Current state-of-the-art steganography techniques are more robust than this, especially in regard to image secrets. Take for instance the most fragile of the steganography techniques used in this work---least significant bit (LSB) method. Changing a localized pixel, even many pixels in-fact, will have a small impact on the secret recovery performance. That is why just adding noise to an image is not effective at sanitizing an image. Text secrets, however, might be this fragile, but this too depends on the length of the message embedded. In regard to JPEG compression, this is a lossy image compression format. There are many cases where a lossless image file format is required, such as medical imaging or legal/forensic imaging.
>
> (R) *The experimental evaluation is poor...There is a lot of prior art in watermark removal. You should show that they do not work.*
>
> (A) If we understood correctly, you are asking why we haven't included works on watermark removal? If so, we have included watermark papers in our literature review (and added a few more as a suggestion from Reviewer 158U). The analysis in Section 2.2 helps to explain our experimental reasoning. For instance, many watermark removal techniques are not blind, as they require information from known hiding techniques. Additionally, while steganography and watermarking are related fields of work, we focus on comparisons to steganography, which only involves hidden information. Watermarking is an interesting field that includes both visible and invisible watermarks. We think comparisons between these two fields will be an interesting area to explore in the future. Thank you for your question!
>
> [1] Hiding images in plain sight: Deep steganography.
>
> [2] Steganographic generative adversarial networks.
>
> [3] An embedding cost learning framework using gan.
>
> [4] Automatic steganographic distortion learning using a generative adversarial network.
>
> [5]  Gan-based steganography with the concatenation of multiple feature maps.
>
> [6] Hidden: Hiding data with deep networks.
>
> [7]  Sstegan: Self-learning steganography based on generative adversarial networks.
>
> [8] Udh:Universal deep hiding for steganography, watermarking, and light field messaging.

---

> > ### Author Response · Authors · 2023-11-21
> > **Reviewer Feedback Cont.**
> >
> > (R) *The capacity of images of size 32x32 (size of images in Cifar 10) will be very small, which means that the experimental settings are distant from the reality.*
> >
> > (A) Thank you for your comment. We agree that DM-SUDS should be evaluated on a more complex dataset. These experiments were included in the original draft of the paper in Section 5, *Evaluation on ImageNet*. In the majority of the experiments, however, we decided to use CIFAR-10 to demonstrate the marked improvement from a previous method, which started to deteriorate with CIFAR-10. We would also like to highlight that many watermarking works utilize CIFAR-10 as a standard dataset [9-15]. This is by no means a comprehensive list.
> >
> > (R) *Have you tried basic JPEG compression and recompression with different quality factors?*
> >
> > (A) Yes, we have tested the sanitization capabilities of JPEG compression. While this works sometimes, we believe that as deep learning steganography continues to progress, it will bypass JPEG compression. We hope that this sanitization technique will extend beyond JPEG compression capabilities, and look forward to addressing this in future work. Additionally, using JPEG compression for a file format is not universal. Due to preference or necessity, lossless formats (png and bitmap) are also used widely. Examples include medical imaging and image forensics for legal processes.
> >
> > (R) *Why you have not tried methods from the famous stirmark test?*
> >
> > (A) Thank you for this suggestion. This is an interesting evaluation for robustness testing, and we appreciate you pointing this out. The norm has not considered this method of testing. We agree that it seems applicable and look forward to evaluating the stirmark in the future.
> >
> > (R) *What is the length of the message you have hidden into the images?*
> >
> > (A) The message we use throughout this work is an image secret. Length, here, is a bit hard to express. The embedded secret images, though, are the same size as the cover image: 3x32x32 for CIFAR-10 and 3x64x64 for ImageNet.
> >
> > [9] Yuki Nagai, Yusuke Uchida, Shigeyuki Sakazawa, and Shin’ichi Satoh. Digital watermarking for
> > deep neural networks.
> >
> > [10] Shichang Sun, Haoqi Wang, Mingfu Xue, Yushu Zhang, Jian Wang, and Weiqiang Liu. Detect
> > and remove watermark in deep neural networks via generative adversarial networks
> >
> > [11] yusuke Uchida, Yuki Nagai, Shigeyuki Sakazawa, and Shin’ichi Satoh. Embedding watermarks
> > into deep neural networks.
> >
> > [12] Peng Yang, Yingjie Lao, and Ping Li. Robust watermarking for deep neural networks via bi-level
> > optimization.
> >
> > [13] Alsharif Abuadbba, Hyoungshick Kim, and Surya Nepal. Deepisign: invisible fragile watermark to
> > protect the integrity and authenticity of cnn
> >
> > [14] Zhaoxia Yin, Heng Yin, and Xinpeng Zhang. Neural network fragile watermarking with no model
> > performance degradation
> >
> > [15] Yingjie Lao, Weijie Zhao, Peng Yang, and Ping Li. Deepauth: A dnn authentication framework
> > by model-unique and fragile signature embedding.

---

> > ### Comment · Reviewer_zZQW · 2023-11-22
> >
> > Thanks for the answer.
> >
> > Steganography is not as volatile as you would expect. In past 10 years, there has been relatively few methods proposed, because it is just hard. I agree that there is a constant flux of papers proposing steganography based on deep learning, but I have not read a paper where they compared to classical methods under Kerckhoffs' settings. They do not even compare to state of the art in the field [1] which uses deep methods to learn the model of images.
> >
> > Modern state of the art steganography is fragile, because there is syndrome coding involved and as I have written previously, ref [2] has shown that it goes directly against robustness. Therefore your method would be intended for the robust steganographic methods, which is niche and close to watermarking, since watermarking (most of the time) needs to be robust. This is why techniques for removing watermark, such as those in old but relevant stirmark test are relevant. You cannot say that these techniques are not relevant and not compare to them. If you say that JPEG recompression does not work, then you should experimentally demonstrate it and show, how many times it fails to remove the message and compare the distortion of your method. The same goes. with other applicable methods in Stirmark test.
> >
> > Plenty of papers say that they are concerned with lossless image formats, but in reality, majority of images on internet are compressed. Companies invest huge amount of money to improve compression and therefore if you do not compress, you are suspicious.
> >
> > Finally, universal steganalysis which is designed to detect any steganographic scheme has been around since 2003 [3] (and there are more modern works but I put old one to show that the problem is known).
> >
> >
> > [1] Bernard, Solène, et al. "Backpack: a Backpropagable Adversarial Embedding Scheme." IEEE Transactions on Information Forensics and Security 17 (2022): 3539-3554.
> >
> > [2] Kin-Cleaves, Christy, and Andrew D. Ker. "Adaptive steganography in the noisy channel with dual-syndrome trellis codes." 2018 IEEE International Workshop on Information Forensics and Security (WIFS). IEEE, 2018.
> >
> > [3] Lyu, Siwei, and Hany Farid. "Steganalysis using color wavelet statistics and one-class support vector machines." Security, steganography, and watermarking of multimedia contents VI. Vol. 5306. SPIE, 2004.

---

> > > ### Author Response · Authors · 2023-11-23
> > > **Reviewer Feedback Pt. 2**
> > >
> > > Thank you for your responses! Reviewer comments are indicated by (R), and our responses are indicated by an (A).
> > >
> > > (R) *Steganography is not as volatile as you would expect. In past 10 years, there has been relatively few methods proposed, because it is just hard. I agree that there is a constant flux of papers proposing steganography based on deep learning, but I have not read a paper where they compared to classical methods under Kerckhoffs' settings. They do not even compare to state of the art in the field [1] which uses deep methods to learn the model of images.*
> > >
> > > (A) Thank you for your comment. This may not be exactly what you were referring to, but please see [4] listed below. We were also not aware of this approach, but it seems interesting given your misgivings related to deep learning steganography. Additionally, the steganographic approaches utilized in this work were not meant to be comprehensive of the entire field but to be representative of the different types of steganography currently available: traditional, dependent deep, and universal deep. Our goal, and what we hope comes across in the paper, is that DM-SUDS works on unseen steganography methods, and that it is a blind approach to sanitization.
> > >
> > > (R) *Modern state of the art steganography is fragile, because there is syndrome coding involved and as I have written previously, ref [2] has shown that it goes directly against robustness. Therefore your method would be intended for the robust steganographic methods, which is niche and close to watermarking, since watermarking (most of the time) needs to be robust. This is why techniques for removing watermark, such as those in old but relevant stirmark test are relevant. You cannot say that these techniques are not relevant and not compare to them. If you say that JPEG recompression does not work, then you should experimentally demonstrate it and show, how many times it fails to remove the message and compare the distortion of your method. The same goes. with other applicable methods in Stirmark test.*
> > >
> > > (A) Thank you for your comment. We addressed our reasoning for the proposed experiments in Section 2.2 of the original paper.
> > >
> > > (R) *Plenty of papers say that they are concerned with lossless image formats, but in reality, majority of images on internet are compressed. Companies invest huge amount of money to improve compression and therefore if you do not compress, you are suspicious.*
> > >
> > > (A) We stand by our decision to utilize lossless image formats and hope to address lossy images in the future. We think this will make an interesting work. Thank you for your suggestion.
> > >
> > > (R) *Finally, universal steganalysis which is designed to detect any steganographic scheme has been around since 2003 [3] (and there are more modern works but I put old one to show that the problem is known).*
> > >
> > > (A) Thank you for referencing this. We think this pairs nicely with the paper you linked in [1]. One of the most interesting aspects of using deep learning for steganographic hiding is that detection algorithms can be utilized in the training process. Deep methods can then be trained to bypass detection methods like [3]. This is very similar to the approach you cited in [1]. Thank you for listing these.
> > >
> > > [1] Bernard, Solène, et al. "Backpack: a Backpropagable Adversarial Embedding Scheme." IEEE Transactions on Information Forensics and Security 17 (2022): 3539-3554.
> > >
> > > [2] Kin-Cleaves, Christy, and Andrew D. Ker. "Adaptive steganography in the noisy channel with dual-syndrome trellis codes." 2018 IEEE International Workshop on Information Forensics and Security (WIFS). IEEE, 2018.
> > >
> > > [3] Lyu, Siwei, and Hany Farid. "Steganalysis using color wavelet statistics and one-class support vector machines." Security, steganography, and watermarking of multimedia contents VI. Vol. 5306. SPIE, 2004.
> > >
> > > [4] https://doi.org/10.1007/s11042-018-6640-y

---

### Official Review · Reviewer_Ha3d · 2023-10-30

**Soundness:** 3 good
**Presentation:** 4 excellent
**Contribution:** 3 good
**Rating:** 8
**Confidence:** 5

**Summary:**

The authors present a blind method for image steganography sanitization that leverages diffusion models. They demonstrate the effectiveness of this method (DM-SUDS) by comparison to SUDS, a prior method for sanitization that leverages a variational autoencoder. Included is an ablation study on the timestep parameter for forward diffusion in their model, and illustrates the necessity of the forward diffusion process and optimal range of timesteps for three types of steganography, including least significant bit (LSB), universal deep hiding (UDH) and dependent deep hiding (DDH) methods.

**Strengths:**

The paper is well written and provides clear study and results that are significantly improved on the prior art (~70+% reduction in MSE , ~20+% improvement in PSNR) with similar elimination of steganography. The idea is novel and makes a lot of sense for appropriate application of diffusion models to steganography sanitization.

**Weaknesses:**

While the method appears highly effective, a more thorough explanation as to how the parameters were selected would improve the paper.  I was expecting some discussion of the strength of the embedding and estimation of the noise present in the images in order to determine the noise variance parameter for the diffusion model. It appears that the noise variance parameter beta is left unspecified and only the time step parameter is identified, with no explanation of how it was set other than the ablation study. Steganography methods often have a strength parameter that can be varied, and it is not clear how important it is to match the diffusion strength to a particular steganography implementation and strength level. Choice of diffusion parameters may be critical to determine the level of steganography that can be sanitized, as evidenced by the ablation study. It appears that the same dataset was used (CIFAR test set) for all of the experiments. If the parameters of the sanitization model (timestep t) were selected based on experiments performed using the same steganography methods and images, the method is technically not completely blind since the diffusion strength was tuned experimentally to the embedding method performance.  The ablation study lacks in quantitative results. The visualization is helpful, but should not replace quantifiable metrics.

**Questions:**

What was the value for beta, and how was timestep parameter t selected?

Page 8, "is a stronger hiding method" -> I would say more robust here, which is directly related to what is being measured (the fragility of the embedding). Stronger steganography implies strength in the steganographic sense and a measure of undetectability - something you are not measuring.

Could the strength of the diffusion be determined by estimating the amount of noise in the images in order to make this method completely blind?

Please provide a table or plot of the performance metrics for the ablation studies.

---

> ### Author Response · Authors · 2023-11-21
> **Reviewer Feedback and Questions**
>
> We would first like to thank you for your review. We have addressed as many of your concerns as possible. Please see our responses below. (R) refers to you, the reviewer, and we have indicated our responses with an (A).
>
> (R) *"It appears that the noise variance parameter beta is left unspecified and only the time step parameter is identified, with no explanation of how it was set other than the ablation study."*
>
> (A)Thank you for pointing this out. This was an oversight on our part. We utilized the cosine beta scheduler from [1], which can be extended to any number of diffusion steps. The beta parameter is dependent on the time step parameters. We have updated this in the paper.
>
> (R) *"Steganography methods often have a strength parameter that can be varied, and it is not clear how important it is to match the diffusion strength to a particular steganography implementation and strength level. Choice of diffusion parameters may be critical to determine the level of steganography that can be sanitized, as evidenced by the ablation study."*
>
> (A) While steganography methods might have different strengths, DM-SUDS does not have to be tuned to a particular type of steganography strength. While we included an ablation study to see if there was a parameter setting where the sanitization capabilities of DM-SUDS breaks, tuning implies that the diffusion step parameter is sensitive to change, which is not the case.
>
> (R)*"It appears that the same dataset was used (CIFAR test set) for all of the experiments. If the parameters of the sanitization model (timestep t) were selected based on experiments performed using the same steganography methods and images, the method is technically not completely blind since the diffusion strength was tuned experimentally to the embedding method performance."*
>
> (A) We also include experiments with ImageNet in Section 5, *Evaluation on ImageNet*. Here we utilize the same arbitrarily chosen timestep value $t=250$ without any tuning.
>
> (R) *What was the value for beta, and how was timestep parameter t selected?*
>
> (A) For the beta value, we utilize a cosine scheduler from [1]. The initial timestep $t$ was arbitrarily chosen. Additional experiments were included regarding timestep $t$ to see if we could a find value where sanitization does not work.
>
> (R) *Page 8, "is a stronger hiding method" -> I would say more robust here, which is directly related to what is being measured (the fragility of the embedding). Stronger steganography implies strength in the steganographic sense and a measure of undetectability - something you are not measuring.*
>
> (A) Thank you for your suggestion! We have updated this in the paper.
>
> (R) *Could the strength of the diffusion be determined by estimating the amount of noise in the images in order to make this method completely blind?*
>
> (A) While this strategy would be useful if we wanted to find a lower bound for $t$ for each element, it is not necessary. We can over-approximate the number of timesteps for all hiding methods, removing this added calculation from the process.
>
> (R) *Please provide a table or plot of the performance metrics for the ablation studies.*
>
> (A) Thank you for this suggestion! We have updated this in the paper.
>
>
> [1] Jonathan Ho, Ajay Jain, and Pieter Abbeel. Denoising diffusion probabilistic models. arXiv preprint
> arxiv:2006.11239, 2020

---

### Meta-Review · Area_Chair_QwYn · 2023-12-05

**Metareview:**

Earlier non-learning steganography has been studied for decades.
In this paper, the authors introduce a blind deep learning steganography sanitization method that utilizes a diffusion model framework to sanitize universal and dependent steganography (DM-SUDS).
Actually, the first round of review ratings are diverse.
For those reviewers with lower ratings, the authors' responses did not satisfy at least two reviewers.
For example, Reviewer 158U points out he/she kept the score the same, and does not think the response convincing.
There are several disagreements between Reviewer zZQW and authors.
An example is that the reviewer disagrees the study of  (DM-SUDS) on lossless images only!
Based on the above concerns, this paper is rejected.

**Justification For Why Not Higher Score:**

The authors' responses did not satisfy at least two reviewers.
For example, Reviewer 158U points out he/she kept the score the same, and does not think the response convincing.
There are several disagreements between Reviewer zZQW and authors.
An example is that the reviewer disagrees the study of  (DM-SUDS) on lossless images only!
Based on the above concerns, this paper is rejected.

**Justification For Why Not Lower Score:**

Partial comments have been address well.

---

### Decision · Program_Chairs · 2024-01-16

Reject